# Finding Landmarks of Covariate Shift with Max-Sliced Kernel Wasserstein Distance

## Abstract

To detect and understand covariate shifts, especially those caused by localized changes in the distribution, we propose a more interpretable divergence through a kernel-based sliced Wasserstein divergence, which is computationally efficient for two-sample testing. The proposed landmark-based slicing seeks a single data point, defining a slice in the reproducing kernel Hilbert space, that maximizes the kernel max-sliced Wasserstein distance. This point and points that surround it from the two samples provide an interpretation of localized divergences. We investigate this new divergence on various shift scenarios and the effect of the choice of learning representations, compared to maximum mean discrepancy (MMD). Results on MNIST and CIFAR-10 dataset demonstrate superior statistical power of the divergence, and analysis of the landmark and its neighborhood are revealing about the discrepancy between the distributions.

## 1 Introduction

Performance of machine learning models are impacted by covariate shift (Quionero-Candela et al., 2009; Shimodaira, 2000), where the distribution of the features (also known as covariates) in the training set do not match the distribution seen in the test set during deployment. Divergence measures quantify the dissimilarity between the distributions (Sriperumbudur et al., 2009) and are essential to detect covariate shift. Previous work Rabanser et al. (2019); Gretton et al. (2009); Jang et al. (2022) has shown that kernel-based maximum mean discrepancy (MMD) (Gretton et al., 2012) provides a tractable and reliable divergence method that can be directly applied to samples. As the performance of such methods depends on the choice of kernel functions, successful covariate shift detection on complicated data such as images requires MMD to be applied on optimized deep kernels (Liu et al., 2020; Cheng & Xie, 2021) or pretrained neural networks representations, such as CLIP (Jayasumana et al., 2024), or classification models' softmax probabilities (Lipton et al., 2018).

Recently, the family of Wasserstein distances (Villani, 2008; Cédric, 2003), which includes the earth mover's distances, has seen renewed interest due to its ability to handle provide a meaningful divergence measure for distributions with disjoint support in terms of the cost of the optimal transport (Chen et al., 2018; Peyré & Cuturi, 2019; Bhatia et al., 2019; Santambrogio, 2015) between the two distributions. While the optimal transport plan underlying the Wasserstein distance is generally challenging to compute, two cases with closed-form solutions are Gaussian distributions and univariate distributions. The latter is exploited by slicing techniques that combine the Wasserstein distance across one-dimensional projections of the distributions yielding computationally tractable divergence measures with statistical advantages (Mueller & Jaakkola, 2015; Bonneel et al., 2015; Kolouri et al., 2019; 2016; Wang et al., 2021). In particular, the sliced Waserstein distance between multivariate distributions integrates the 1D Wasserstein distance after rank-1 projection defined by a slice across all possible slices, as in the Radon transform (Rabin et al., 2012; Bonneel et al., 2015; Kolouri et al., 2016) Practically, it is estimated from random slices using Monte Carlo integration. As many of the slices may not yield useful divergences, the max-sliced Wasserstein (Deshpande et al., 2019) replaces the random projections with a single slice or an optimized slice distribution (Nguyen et al., 2021) that maximizes the distance under a diversity constraint; however, finding the optimal slice is hard and

previous work relies on relaxation to achieve an approximation (Mueller & Jaakkola, 2015; Wang et al., 2022).

The slice/projection that maximizes the Wasserstein distance be interpreted as defining a linear 'witness function' that reveals the maximal discrepancy between two distributions.[1] This is analogous to MMD's witness, a linear slice in the kernel-induced Hilbert space (Gretton et al., 2012), where points with large magnitude witness function evaluations associated to the maximal discrepancy between unnormalized kernel density estimates, enabling post-hoc interpretation of the divergence.

Similarly, the Kullback-Leibler and other $f$-divergences (Kullback & Leibler, 1951; Ali & Silvey, 1966; Rényi, 1961), are also also interpretable, in the sense that points with the largest magnitude log density ratio are associated with discrepancies. However, the density functions are unknown, the estimation of the underlying densities is challenging (Vapnik, 2000), and while variational optimizations (Nguyen et al., 2010; Rhodes et al., 2020) can provide estimates with bounded error when $f$ is strongly convex (Verine et al., 2023), the $f$-divergences require one distribution to be supported on the support of the second.[2]

One potential drawback of the interpretation of the witness function for the max-sliced Wasserstein distance and MMD (or the log-density ratio in f-divergences) is that the witness function is generally not localized, meaning that points with high witness function evaluations may come from disparate discrepancies spread throughout the space. Motivated by cases where the covariate shift itself is caused by under or over-representation in a localized subset of the distribution, such as a mode, we propose an approach called **max-sliced kernel landmark Wasserstein distance (MLW)** that ensures a localized witness function based on the kernel embedding of a single point (the landmark), which acts as a non-linear slicing function, to identify the maximal localized discrepancy between two distributions or datasets. As in kernel-based Wasserstein distance (Zhang et al., 2020; Oh et al., 2020), the data points are mapped to a Hilbert space, and divergence is measured as the Wasserstein distance between the distributions of the landmark-induced witness function evaluations. The landmark point, defining the witness function, is chosen to maximize this divergence, and it along with its neighbors are easily interpretable as where the distributions most differ; kernel-based Wasserstein distance does not provide such interpretation. Importantly, landmark-based kernel max-slicing has a closed-form solution for two samples with at most $n$ points that requires $\mathcal{O}(n^2 \log(n))$ operations compared to kernel max-slicing (Brockmeier et al., 2021; Wang et al., 2022; 2024), which is a NP-Hard (Wang et al., 2024) problem requiring approximation through iterative or a semi-definite relaxation. The efficiency of MLW can be used to form two types of two-sample tests through permutation test (Good, 2013) that generates a surrogate distribution under the null hypothesis. In the first case, the divergence is calculated on each of the $B$ permutations requiring $\mathcal{O}(n^2 \log(n) + Bn^2 + B\log(n)) = \mathcal{O}(n^2 \log(n) + Bn^2)$ operations, in comparison MMD requires $\mathcal{O}(Bn^2)$. In the second case, the samples are split with a fixed proportion: the first split is used to calculate the landmark, and the second split is used to calculate the test statistic and the surrogate distribution, which only requires $\mathcal{O}(n \log(n) + Bn)$ operations, which is computationally cheaper than MMD.

MLW shares motivation with an earlier interpretable two-sample test: the mean embedding (ME) divergence (Chwialkowski et al., 2015). The ME algorithm optimizes the location of inducing points, which are not in the sample but act as landmarks, in order to maximize a mean embedding test statistic (Chwialkowski et al., 2015). The contribution of MLW is that an interpretable test can be obtained with closed-form optimization by considering just the sample data points as landmarks and using the Wasserstein distance for a single landmark to obtain increased power. Optimizing multiple landmarks was concerned in the context of model criticism (Kim et al., 2016), where a set of prototype points are used to approximate a distribution's kernel mean embedding (the two samples are then the set of prototype and the full dataset, respectively). Kim et al. (2016) select landmarks greedily based on the largest magnitude witness function evaluations, with a log-determinant anti-concentration penalty. The first landmark is then the MMD baseline.

Following previous work (Rabanser et al., 2019; Wang et al., 2022), we test our methods ability to detect covariate shifts caused by class imbalance. This is practically relevant to understand if an unlabeled sample

---

[1]While not specifically divergences, there are other methods to find a projection that distinguishes distributions using sufficient dimensional reduction (Meng et al., 2019; Li, 1991).

[2]Interestingly, the earth mover's distance and other integral probability metrics (Sriperumbudur et al., 2009) can be combined with $f$-divergences through infimal convolution (Farnia & Tse, 2018; Birrell et al., 2022), allowing disjoint support.

of points, such as a test set, has the same distribution as the training set. The classes can also serve as proxy for detecting changes in the prevalence of 'modes' of a distribution. In the case of class imbalance, we create powerful detection methods by combining MLW with either pretrained CLIP embeddings (Radford et al., 2021) or Black Box Shift Detection (BBSD) (Lipton et al., 2018), which uses the trained classifier as the representation to detect whether the source (training) and target (test) distributions are the same. Results on synthetically imbalanced real-world datasets show that this landmark-based max slicing is statistically more powerful than maximum mean discrepancy (MMD) (Gretton et al., 2012). Beyond standard MMD, we benchmark MLW against other methods that optimize the form of the test statistic using split samples, including kernel-based approaches (Liu et al., 2020; Cheng & Xie, 2021; Wang et al., 2022; 2024) and the projected Wasserstein distance (Wang et al., 2021). MLW is a notably simpler approach that outperforms these methods on most cases.

We also show how the divergence is interpretable by presenting the landmark along with its neighbors and/or those most dissimilar to it. This can be quantitatively evaluated in experiments by calculating the precision as the proportion of the neighbors that are in fact from imbalanced class/modes.

## 2 Preliminaries and Prior Work

### 2.1 Notation and Definitions

Vectors are represented as boldface $\mathbf{x}$; $\mathbf{1}_n = [1, \ldots, 1]^\top$ is a vector of $n$ ones. Elementwise operations are denoted as follows: product as $\mathbf{x} \circ \mathbf{y}$, $p$th power as $\mathbf{x}^{\circ p}$, and absolute value as $|\mathbf{x}|$. Transpose is denoted $\mathbf{x}^\top$. Sets are represented as calligraphic font like $\mathcal{X}$, as doublestruck letters like $\mathbb{R}$, or, for the first $n$ natural numbers as $[n] = \{1, \ldots, n\}$. $\mathcal{X}^n = \mathcal{X} \times \cdots \times \mathcal{X}$ denotes $n$ repeated Cartesian products. Keeping with this notation the probability simplex is the set of non-negative vectors of dimension $n$ whose entries sum to one, and is denoted $\mathcal{P}_n = \{\mathbf{p} \in [0,1]^n : \sum_{i=1}^n p_i = 1\} \subset \mathbb{R}^n$ (This set is often denoted $\Delta^n$ or $\Delta^{n-1}$). A permutation $\pi$ on $[n]$ is a bijective function $\pi : [n] \to [n]$ that can be stored as vector $\boldsymbol{\pi}$ with entries $\pi_i = \pi(i), \forall i \in [n]$; the set of permutation vectors of length-$n$ is denoted $\mathcal{Q}_n$. $\mathbb{S}^{d-1} \subset \mathbb{R}^d$ denotes the $d$-dimensional unit hypersphere such that $\mathbf{x} \in \mathbb{S}^{d-1} \implies \|\mathbf{x}\|_2 = 1$.

We consider two random variables $X, Y \in \mathcal{X}$ in a feature domain $\mathcal{X}$, distributed according to $X \sim \mu$ and $Y \sim \nu$, with $\mu, \nu \in \mathcal{P}_\mathcal{X}$, where $\mathcal{P}_\mathcal{X}$ is the set of Borel probability measures on the metric space $(\mathcal{X}, \mathfrak{d})$ defined by the distance metric $\mathfrak{d}(x, y), \forall x, y \in \mathcal{X}$. When $\mathcal{X} \subseteq \mathbb{R}^d$, we restrict our attention to those with finite moments. Let $\mathcal{P}_p(\mathbb{R}^d)$ denote the set of probability measures with finite $p$th moments. For a random vector $\mathbf{x} \sim \mu$, $d > 1$, $\mathbb{E}[\|\mathbf{x}\|_2] < \infty$ implies $\mu \in \mathcal{P}_2(\mathbb{R}^d)$, where $\|\cdot\|_2$ denotes the Euclidean norm. For $d = 1$, let $F_X(x) = \mu((-\infty, x])$ and $F_X^{-1}(u) = \inf\{x \in \mathbb{R} : F_X(x) \geq u\}$ denote the cumulative distribution function and its inverse for $X \sim \mu$. When $\mathcal{X}$ is discrete or one of the measures are discrete, then the probability measure can be expressed as a probability mass function $\mu(z) = \sum_{x \in \text{supp}(\mu)} \mu(x) \delta_{\{x\}}(z)$, where $\delta_\mathcal{Z}(z) = \begin{cases} 1, & z \in \mathcal{Z} \\ 0, & z \notin \mathcal{Z} \end{cases}$ is an indicator function. When the support is finite $\text{supp}(\mu) = \{x_i\}_{i=1}^m$, then $\mu(z) = \sum_{i=1}^m \mu_i \delta_{\{x_i\}}(z)$ where $\mu_i = \mu(x_i), \forall i \in [m]$ and $\boldsymbol{\mu} \in \mathcal{P}_m$. A finite discrete distributions $\hat{\mu} = \sum_{i=1}^n \mu_i \delta_{\{x_i\}}$ corresponds to a weighted sample $\{(\mu_i, x_i)\}_{i=1}^n \in \mathcal{P}_n \times \mathcal{X}^n$.

### 2.2 Optimal Transport Based Divergences

The optimal transport (OT) problem underlying the Wasserstein distance depends on the distance $\mathfrak{d}$ function defining the metric space. The choice of Euclidean metric $\mathfrak{d}_\mathrm{E}(\mathbf{x}, \mathbf{y}) = \|\mathbf{x} - \mathbf{y}\|_2$ for distributions supported on $\mathbb{R}^d$, corresponds to the earth mover's distance (EMD) $\inf_{\gamma \in \mathcal{P}_{\mu,\nu}} \mathbb{E}_{(X,Y) \sim \gamma}[\mathfrak{d}(X, Y)]$, where $\mathcal{P}_{\mu,\nu}$ is the set of joint distributions with marginals $\mu$ and $\nu$. More generally the Wasserstein-$p$ distance can be defined for any metric space $(\mathcal{X}, \mathfrak{d})$ and $p \geq 1$—assuming finite $p$th moments of $\mathfrak{d}(X, Y)$—as

$$W_{\mathfrak{d}^p}(\mu, \nu) = \left( \inf_{\gamma \in \mathcal{P}_{\mu,\nu}} \mathbb{E}_{(X,Y) \sim \gamma}[\mathfrak{d}(X, Y)^p] \right)^{\frac{1}{p}}. \tag{1}$$

$W_{\mathfrak{d}^p}(\mu, \nu)$ is a metric between probability distributions in $\mathcal{P}_{\mathcal{X}}$ (Cédric, 2003; Villani, 2008; Peyré & Cuturi, 2019). Two important cases are EMD $W^1_{\mathfrak{d}^1_E}(\mu, \nu) = W_{\mathfrak{d}_E}(\mu, \nu)$ and $W^2_{\mathfrak{d}^2_E}(\mu, \nu)$.

### 2.2.1 Optimal Transport for Discrete Measures

When both distributions are finite discrete measures $\hat{\mu} = \sum_{i=1}^m \mu_i \delta_{\{x_i\}}$ and $\hat{\nu} = \sum_{i=1}^n \nu_i \delta_{\{y_i\}}$ as is the case with data sets, $\{(\mu_i, x_i)\}_{i=1}^m \in \mathcal{P}_m \times \mathcal{X}^m$ and $\{(\nu_i, y_i)\}_{i=1}^n \in \mathcal{P}_n \times \mathcal{X}^n$, then the Wasserstein distance requires solving the linear program

$$W^p_{\mathfrak{d}^p}(\hat{\mu}, \hat{\nu}) = \min_{\mathbf{P} \in \mathcal{P}_{\boldsymbol{\mu},\boldsymbol{\nu}}} \sum_{i=1}^m \sum_{j=1}^n P_{ij} C_{ij}, \tag{2}$$

where $C_{ij} = \mathfrak{d}^p(x_i, y_j)$ and the transport plan $\hat{\gamma} = \sum_{i=1}^m \sum_{j=1}^n P_{ij} \delta_{\{(x_i, y_j)\}}$ is expressed in terms of a discrete joint distribution $\mathbf{P} \in \mathcal{P}_{\boldsymbol{\mu},\boldsymbol{\nu}}$ coupling the probability masses $\boldsymbol{\mu} = [\mu_i]_{i=1}^m$ and $\boldsymbol{\nu} = [\nu_i]_{i=1}^n$, where $\mathcal{P}_{\boldsymbol{\mu},\boldsymbol{\nu}} = \{\mathbf{P} \in \mathcal{P}_m \times \mathcal{P}_n : \mathbf{P}\mathbf{1}_n = \boldsymbol{\mu} \text{ and } \mathbf{P}^\top \mathbf{1}_m = \boldsymbol{\nu}\}$ is defined by the set of linear equality constraints ensuring the marginals of the joint match $\boldsymbol{\mu}$ and $\boldsymbol{\nu}$. The computational complexity of algorithms to solve this linear program are $\mathcal{O}(n^3)$ assuming $n \geq m$ (Peyré & Cuturi, 2019).

### 2.2.2 One-dimensional Optimal Transport

The optimal transport between one-dimensional (1D) distributions $\mathcal{X} = \mathbb{R}$ with $\mathfrak{d}(x, y) = \mathfrak{d}_E(x, y) = |x - y|$ is solved analytically in terms of the inverse cumulative distribution functions (Santambrogio, 2015),

$$W^p_{\mathfrak{d}^p_E}(\mu, \nu) = \inf_{\gamma \in \mathcal{P}_{\mu,\nu}} \mathbb{E}_{(X,Y)\sim\gamma} \left[|X - Y|^p\right] = \int_0^1 |F_X^{-1}(u) - F_Y^{-1}(u)|^p du. \tag{3}$$

(For $p = 1$, EMD can be calculated from the cumulative distributions $W_{\mathfrak{d}_E}(\mu, \nu) = \int_{-\infty}^{\infty} |F_X(u) - F_Y(u)| du$.)

### 2.2.3 1D Optimal Transport for Discrete Measures

In the case of finite discrete measures corresponding to samples of 1D points, the solution to the linear program, which is equivalent to the integral of the $p$th power of the differences between the inverse cumulative distributions, can be computed efficiently after sorting, which requires only $\mathcal{O}(n \log(n))$ operations, and avoids the quadratic cost of computing the distances between all pairs of data points. Let $\boldsymbol{\pi} \in \mathcal{Q}_m$ and $\boldsymbol{\sigma} \in \mathcal{Q}_n$ denote the permutations such that $x_{\pi_1} \leq x_{\pi_2} \leq \cdots \leq x_{\pi_m}$ and $y_{\sigma_1} \leq y_{\sigma_2} \leq \cdots \leq y_{\sigma_m}$, respectively, and let $\acute{\mathbf{x}} = [x_{\pi_i}]_{i=1}^m$ and $\acute{\mathbf{y}} = [y_{\sigma_i}]_{i=1}^n$ denote the sorted samples.

In the case of equal-sized samples $m = n$ with uniform masses $\boldsymbol{\mu} = \boldsymbol{\nu} = \frac{1}{m}\mathbf{1}_m$, $W^p_{\mathfrak{d}^p_E}(\hat{\mu}, \hat{\nu}) = \frac{1}{m} \sum_{i=1}^m |x_{\pi_i} - y_{\sigma_j}|^p = \frac{1}{m}\|\acute{\mathbf{x}} - \acute{\mathbf{y}}\|_p^p$. For general masses $\boldsymbol{\mu}$ and $\boldsymbol{\nu}$, an optimal transport plan matrix can be found after sorting in $\mathcal{O}(n)$ operations by the 'Northwest corner rule' dating back to Dantzig (Charnes & Cooper, 1954; Hoffman et al., 1963). (If $p > 1$ and there are no duplicate points, then the Northwest corner rule solution is a unique solution (Dubuc et al., 1999).)

### 2.2.4 Sliced Optimal Transport

The computational efficiency of sorting versus solving a linear program has motivated multiple approaches for mapping vector-valued data in an Euclidean space to 1D subspaces via rank-1 projections—a process known as slicing, computing 1D Wasserstein distances in each subspace, and aggregating the distances across slices. While the aggregation is formally expressed as integrals, in practice, it is estimated by Monte Carlo integration, averaging over a finite number of slices defined by random unit vectors sampled from the unit hypersphere to compute a sliced Wasserstein distance (Rabin et al., 2012; Bonneel et al., 2015; Wu et al., 2019; Deshpande et al., 2018; Kolouri et al., 2017). Alternatively, max-sliced Wasserstein (Deshpande et al., 2019) seeks the single slice that maximizes the 1D divergence. Other variants include generalized slicing (Kolouri et al., 2019), distributional slice Wasserstein distance (Nguyen et al., 2021), and energy-based sliced Wasserstein (Nguyen & Ho, 2023).

In the context of the Euclidean distance, a slice is defined by a unit vector $\mathbf{w} \in \mathbb{S}^{d-1}$ such that $\mathbf{w}\mathbf{w}^\top \in \mathbb{R}^{d \times d}$ is a rank-1 projection matrix defining a subspace. The distance in this subspace $\mathfrak{d}_\mathbf{w}$ is defined as

$$\mathfrak{d}_\mathbf{w}(\mathbf{x}, \mathbf{y}) = \|(\mathbf{w}\mathbf{w}^\top)(\mathbf{x} - \mathbf{y})\|_2 = \|\mathbf{w}\|_2 |\mathbf{w}^\top(\mathbf{x} - \mathbf{y})| = |\mathbf{w}^\top \mathbf{x} - \mathbf{w}^\top \mathbf{y}| = \mathfrak{d}_\mathrm{E}(\mathbf{w}^\top \mathbf{x}, \mathbf{w}^\top \mathbf{y}).$$

Let $\theta(\mathbf{x}) = \mathbf{w}^\top \mathbf{x}$ denote the slicing as a function. The Wasserstein-$p$ distance in this subspace is equivalent to computing Wasserstein-$p$ distance between the pushforward measures $\theta_\sharp \mu$ and $\theta_\sharp \nu$, $W^p_{\mathfrak{d}_\mathbf{w}}(\mu, \nu) = W^p_{\mathfrak{d}^p_\mathrm{E}}(\theta_\sharp \mu, \theta_\sharp \nu)$. For finite discrete measures with $\boldsymbol{\mu} = \boldsymbol{\nu} = \frac{1}{m}\mathbf{1}_m$, $W^p_{\mathfrak{d}_\mathbf{w}}(\hat{\mu}, \hat{\nu}) = \frac{1}{m}\sum_{j=1}^{m}|\mathbf{w}^\top \mathbf{x}_{\pi_j} - \mathbf{w}^\top \mathbf{y}_{\sigma_j}|^p$, where the permutations $\boldsymbol{\pi}$ and $\boldsymbol{\sigma}$ ensure that $\mathbf{w}^\top \mathbf{x}_{\pi_1} \leq \mathbf{w}^\top \mathbf{x}_{\pi_2} \leq \cdots \leq \mathbf{w}^\top \mathbf{x}_{\pi_m}$ and $\mathbf{w}^\top \mathbf{y}_{\sigma_1} \leq \mathbf{w}^\top \mathbf{y}_{\sigma_2} \leq \cdots \leq \mathbf{w}^\top \mathbf{y}_{\sigma_m}$, respectively. The general case of masses requires computing the transport solution via the Northwest corner rule given the permuted masses.

The sliced Wasserstein distance is defined as the expectation across slices $\xi \in \mathcal{P}_{\mathbb{S}^{d-1}}$ where $\xi = \mathrm{Uniform}(\mathbb{S}^{d-1})$ is a uniform distribution over the hypersphere, $SW^p_p(\mu, \nu) = \mathbb{E}_{\mathbf{w} \sim \xi}[W^p_{\mathfrak{d}_\mathbf{w}}(\mu, \nu)]$. The max-sliced Wasserstein-$p$ distance is $MSW_p(\mu, \nu) = \sup_{\mathbf{w} \in \mathbb{S}^{d-1}} W^p_{\mathfrak{d}_\mathbf{w}}(\mu, \nu)$. The max-sliced Wasserstein-$p$ distance seeks a single subspace that can maximally distinguish the distributions. Clearly, this is a probability metric since the search encompasses all possible one-dimensional projections and $MSW_p$ inherits the probability metric properties of the Wasserstein-$p$ distance. However, in practice obtaining the max-slice is itself a difficult optimization even for discrete measures.

## 2.3 Kernel Based Divergences

The kernel trick is a well-studied approach to create non-linear functions as linear parametrizations of data embedded in a reproducing kernel Hilbert space (RKHS). Given a symmetric (real-valued) positive semi-definite kernel function $\kappa : \mathcal{X} \times \mathcal{X} \to \mathbb{R}$, such that for any sample size $n$ and sample of points $\{x_i\}_{i=1}^n \subseteq \mathcal{X}^n$, and scalars $c_1, \cdots, c_n \in \mathbb{R}$, $\sum_{i=1}^n \sum_{j=1}^n c_i c_j \kappa(x_i, x_j) \geq 0$, then $\kappa$ uniquely defines a reproducing kernel Hilbert space $\mathcal{H}$, with inner products denoted $\langle \cdot, \cdot \rangle$, and satisfies two key properties (Scholkopf & Smola, 2001): $\forall x \in \mathcal{X}, \quad \kappa(\cdot, x) \in \mathcal{H}$ and $\forall x \in \mathcal{X}, h \in \mathcal{H}, \quad h(x) = \langle h, \kappa(\cdot, x) \rangle$. The second is the reproducing property. The properties can also expressed in terms of embedding function $\phi : \mathcal{X} \to \mathcal{H}$: 1) $\forall x \in \mathcal{X}$ $\phi(x) = \kappa(\cdot, x) \in \mathcal{H}$, and 2) $\forall h \in \mathcal{H}, h(x) = \langle h, \phi(x) \rangle$. Additionally, $\forall x, y \in \mathcal{X}, \langle \phi(x), \phi(y) \rangle = \kappa(x, y)$.

### 2.3.1 Maximum Mean Discrepancy

In terms of statistical divergences, if the kernel obeys certain properties, referred to as being characteristic (Fukumizu et al., 2008), then the embedding of a distribution $\mu$ in the Hilbert space $\bar{h}(\mu) = \mathbb{E}_{X \sim \mu}[\phi(X)] = \bar{\phi}_X$ will completely characterize it. Additionally, we assume a bounded kernel $\mathbb{E}_{X \sim \xi}[\kappa(X, X)] \leq \infty, \quad \forall \xi \in \mathcal{P}_\mathcal{X}$. In these cases, the mean embedding function $\bar{h} : \mathcal{P}_\mathcal{X} \to \mathcal{H}$ is injective (Fukumizu et al., 2008), and it is sufficient to distinguish two distributions by the norm of the difference of their mean embeddings as in maximum mean discrepancy (MMD) (Gretton et al., 2009) : $MMD_\kappa(\mu, \nu) = \|\bar{h}(\mu) - \bar{h}(\nu)\|_\mathcal{H}$. MMD is interpretable as it can be posed as an optimization of the witness function as a functional slice $\omega \in \mathcal{F} = \{\omega \in \mathcal{H} : \|\omega\|_\mathcal{H} \leq 1\}$, which due to the reproducing property, is a linear slice in the RKHS, to maximize the differences in means,

$$MMD_\kappa(\mu, \nu) = \sup_{\omega \in \mathcal{F}} \mathbb{E}_{X \sim \mu, Y \sim \nu}[\langle \phi(X) - \phi(Y), \omega \rangle] = \sup_{\omega \in \mathcal{F}} \langle \bar{\phi}_X - \bar{\phi}_Y, \omega \rangle = \sup_{\omega \in \mathcal{F}} \mathbb{E}[\omega(X) - \omega(Y)], \qquad (4)$$

which has the solution $\omega^* = \frac{\bar{\phi}_X - \bar{\phi}_Y}{\|\bar{\phi}_X - \bar{\phi}_Y\|_\mathcal{H}}$. Intuitively, the optimal witness function is the difference of unnormalized kernel density estimates under the two distributions: $\omega^*(x) \propto \hat{p_X}(x) - \hat{p_Y}(x)$ with $\hat{p_X}(x) = \mathbb{E}_{X \sim \mu}[\kappa(x, X)]$ and $\hat{p_Y}(x) = \mathbb{E}_{Y \sim \nu}[\kappa(x, Y)]$. In the discrete case, the objective is $\mathbb{E}[\omega(X) - \omega(Y)] = \sum_{z \in \mathcal{X}}(\mu(z) - \nu(z))\omega(z)$, and it is clear that the $\omega^*(z)$ is positive value when $\mu(z) > \nu(z)$ and negative when $\nu(z) > \mu(z)$. Furthermore, the largest magnitude evaluations of $\omega^*$ indicate the points where $\mu$ and $\nu$ differ the most, allowing for interpretation of the discrepancies.

MMD can be readily estimated from the kernel evaluations $MMD^2_\kappa(\mu, \nu) = \mathbb{E}[\kappa(X, X')] - 2\mathbb{E}[\kappa(X, Y)] + \mathbb{E}[\kappa(Y, Y')]$, where expectations are taken over pairs of two independent copies of the four independent random variables $X \sim \mu$, $X' \sim \mu$, $Y \sim \nu$, and $Y' \sim \nu$. For finite discrete measures, the empirical

witness function $\hat{\omega}^*(z) = \frac{1}{MMD_\kappa(\hat{\mu},\hat{\nu})} \left( \sum_{i=1}^m \mu_i \kappa(x_i, z) - \sum_{i=1}^n \nu_i \kappa(y_i, z) \right)$, corresponds to the biased estimate $MMD_\kappa^2(\hat{\mu}, \hat{\nu}) = \sum_{i=1}^m \sum_{j=1}^m \mu_i \mu_j \kappa(x_i, x_j) - 2 \sum_{i=1}^m \sum_{j=1}^n \mu_i \nu_j \kappa(x_i, y_j) + \sum_{i=1}^n \sum_{j=1}^n \nu_i \nu_j \kappa(y_i, y_j)$ of MMD.

### 2.3.2  Kernel Wasserstein Distance

Kernel methods induce metric spaces—a useful property in cases where data does not have a natural vector representation. Depending on the domain $\mathcal{X}$, a kernel-induced distance may be more appropriate than other distance metrics and can enable Wasserstein distances in spaces without a natural distance metric. That is, the kernel-induced distance $\mathfrak{d}_\kappa(x, y) = \|\phi(x) - \phi(y)\|_{\mathcal{H}} = \kappa(x, x) - 2\kappa(x, y) + \kappa(y, y)$ can be used to compute a kernel Wasserstein-$p$ distance (Zhang et al., 2020), denoted as $W_{\mathfrak{d}_\kappa^p}(\mu, \nu)$. The kernel Wasserstein-2 distance can be related to MMD, by the inequality $MMD(\mu, \nu) \leq W_{\mathfrak{d}_\kappa^2}(\mu, \nu)$, which follows from the fact that the MMD is a lower bound on the kernel Bures-Wasserstein divergence (Oh et al., 2020; Zhang et al., 2020), and the Bures-Wasserstein divergence lower bounds the Wasserstein-$p$ distance (Gelbrich, 1990). The Bures-Wasserstein divergence is also known as the Fréchet distance, and in the kernel case it is the combination of difference of kernel means (MMD) and kernel variances. For finite samples of size $n$, the computational complexity of the kernel divergences are as follows: MMD's complexity is $\mathcal{O}(n^2)$, kernel Bures-Wasserstein divergence is $\mathcal{O}(n^3)$ due to computation of matrix square root (Oh et al., 2020), and the kernel Wasserstein distance is $\mathcal{O}(n^3)$. Another kernel-based approach is the kernel Fisher discriminant ratio (KFDR) (Harchaoui et al., 2008), which requires matrix inversion and has computational complexity of $\mathcal{O}(n^3)$. Thus, MMD is the most scalable method.

### 2.3.3  Max-sliced Kernel Wasserstein Distance

To make a kernel Wasserstein distance-based divergence that is more efficient and interpretable, one can consider slicing in the RKHS, leading to sliced kernel Wasserstein distances applicable to random variables without vector-space representations where the linear slices in the RKHS are non-linear witness functions in the input space.

Mathematically, a witness function $\omega \in \{\omega \in \mathcal{H} : \quad \|\omega\|_{\mathcal{H}} = 1\} \subset \mathcal{F}$ defines a projection operator to a one-dimensional subspace $\omega \otimes \omega \in \mathcal{H} \times \mathcal{H}$. The sliced kernel-induced distance is $\mathfrak{d}_\omega(x, y) = \|(\omega \otimes \omega)(\phi(x) - \phi(y))\|_{\mathcal{H}} = \|\omega\|_{\mathcal{H}} |\langle \omega, \phi(x) - \phi(y)| = |\omega(x) - \omega(y)|$, where $\omega(x)$ and $\omega(x)$ are real-valued function evaluations of the slicing function. A kernelized $MSW_p$, the max-sliced kernel Wasserstein distance (Wang et al., 2022; 2024), can be expressed in terms of witness functions as $MSKW_{p,\kappa}(\mu, \nu) = \sup_{\omega \in \mathcal{F}} W_{\mathfrak{d}_\omega^p}(\mu, \nu)$. Intuitively, optimizing the witness function to maximize the Wasserstein distance of the pushforward distribution of the witness function evaluations should be more powerful than only comparing just the means as in MMD. However, unlike MMD there is closed-form solution for the optimal witness function. For finite and discrete measures a representer theorem, along the lines of the kernel projected Wasserstein distance (Wang et al., 2022), which generalizes the representer theorem for regularized empirical risk (Schölkopf et al., 2001) to the constrained case, can be stated (Proof is in Appendix A.1.)

**Theorem 1.** *For two empirical measures $\hat{\mu}$ and $\hat{\nu}$ supported on the pooled set of data points $\{z_i\}_{i=1}^{m+n}$, the optimal slice $\omega^* \in \mathcal{S} = \mathrm{span}(\{\phi(z_i)\}_{i=1}^{m+n}) \cap \mathcal{F} \subset \mathcal{F}$, can be parametrized as linear combination of the pooled data point embeddings $\omega^* = \sum_{i=1}^{m+n} \alpha_i \phi(z_i)$, such that $\forall \omega \in \mathcal{F}$, $MSKW_{p,\kappa}(\hat{\mu}, \hat{\nu}) = \max_{\omega_{\mathcal{S}} \in \mathcal{S}} W_{\mathfrak{d}_{\omega_{\mathcal{S}}}^p}(\mu, \nu) = W_{\mathfrak{d}_{\omega^*}^p}(\mu, \nu) \geq W_{\mathfrak{d}_\omega^p}(\mu, \nu)$.*

However, this is a difficult saddlepoint optimization (NP-Hard (Wang et al., 2024)) as in the original max-sliced Wasserstein distance (Deshpande et al., 2019), with approximations through semidefinite relaxations (Mueller & Jaakkola, 2015; Wang et al., 2022; 2024). Additionally, like MMD the witness function will not be localized.

## 3  Proposed Methodology

To achieve an efficient and interpretable kernel-sliced Wasserstein distance, we propose to limit the witness function to those defined by the Hilbert-space embedding of the sample data points. As the data points that have maximal divergence are useful guides in identifying localized discrepancies, we refer to them as

landmarks.[3] The algorithm to compute the distance and find the landmark is efficient, with a computational complexity of $\mathcal{O}(n^2 \log(n))$ for samples of size $n$, enabling two-sample testing with independent and identically distributed data points in each sample via a permutation test that calculates a surrogate distribution under the null hypothesis by computing the proposed divergence after randomly shuffling data points between the samples.

### 3.1 Landmark Max-Sliced Kernel Wasserstein Distance

To ensure witness functions are unit-norm, we restrict our attention to the witness functions corresponding to the embeddings from normalized, shift-invariant kernels $\kappa(x, y) = \langle \phi(x), \phi(y) \rangle$ such that $\|\phi(z)\|_{\mathcal{H}} = 1 \quad \forall z \in \mathcal{X}$. We refer to a point $z \in \mathcal{X}$ defining the witness function acting as a slice in the RKHS $\omega = \phi(z) \in \mathcal{H}$ as a **landmark**. The landmark kernel-induced distance, $\mathfrak{d}_{\phi(z)}(x, y) = \mathfrak{d}_{\mathrm{E}}(\kappa(z, x), \kappa(z, y)) = |\kappa(z, x) - \kappa(z, y)|$ compares the points $x, y$ in terms of their kernel similarity to the landmark $z$. Given $\mu, \nu \in \mathcal{X}$, we denote the set of landmarks as $\mathcal{L} = \{\phi(z) \in \mathcal{H} : z \in \mathcal{X}\} \subset \{\omega \in \mathcal{H} : \|\omega\|_{\mathcal{H}} = 1\}$. As a constrained version of the max-sliced kernel Wasserstein distance (Brockmeier et al., 2021; Wang et al., 2022; 2024), we propose the landmark max-sliced kernel Wasserstein distance

$$MLW_{\kappa,p}(\mu, \nu) = \sup_{\omega \in \mathcal{L}} W_{\mathfrak{d}_\omega^p}(\mu, \nu) = \sup_{z \in \mathcal{X}} W_{\mathfrak{d}_{\phi(z)}^p}(\mu, \nu) \leq MSKW_{p,\kappa}(\mu, \nu). \tag{5}$$

**Theorem 2.** *If $\kappa$ is characteristic (Fukumizu et al., 2008) (or universal (Micchelli et al., 2006)) and normalized $\kappa(z, z) = 1 \quad \forall z \in \mathcal{X}$, then $MLW_{\kappa,p}(\mu, \nu)$ is a probability distance metric for $\mu, \nu, \xi \in P(\mathcal{X})$ with finite pth moments.*

The proof is in Appendix A.1. While Theorem 2 demonstrates that seeking landmark slices across the whole space yields a probability metric, in practice for finite discrete measures, the landmarks are restricted to points in the support $\mathcal{Z} = \mathrm{supp}(\hat{\mu}) \cup \mathrm{supp}(\hat{\nu}) \subseteq \mathcal{X}$. We define the discrete case as

$$MLW_{\kappa,p}(\hat{\mu}, \hat{\nu}) = \max_{z \in \mathcal{Z}} W_{\mathfrak{d}_{\phi(z)}^p}(\mu, \nu), \tag{6}$$

and note that $MLW_{\kappa,p}(\hat{\mu}, \hat{\nu}) \leq \sup_{z \in \mathcal{X}} W_{\mathfrak{d}_{\phi(z)}^p}(\mu, \nu) \leq MSKW_{p,\kappa}(\hat{\mu}, \hat{\nu})$. However, we note that $MLW_{\kappa,p}(\hat{\mu}, \hat{\nu}) = 0 \implies MSKW_{p,\kappa}(\hat{\mu}, \hat{\nu}) = 0$, and likewise, in contrapostive form $MMD(\hat{\mu}, \hat{\nu}) > 0 \implies MLW_{\kappa,p}(\hat{\mu}, \hat{\nu}) > 0$. Proofs follow along the same lines as those from Theorem 2 combined with Theorem 1.

#### 3.1.1 Efficient Computation of MLW

For finite discrete measures, the main computational cost of landmark max-sliced kernel Wasserstein distance is the sorting of the entries for each slice. The maximization over kernel landmark slices is a simple discrete optimization over the set of $l = m + n$ possible landmarks $\mathcal{Z} = \{z_i\}_{i=1}^l = \{x_i\}_{i=1}^m \cup \{y_i\}_{i=1}^n$. For $k \in [l]$, let $\boldsymbol{\pi}^k$ and $\boldsymbol{\sigma}^k$ denote the permutations such that $\kappa(x_{\pi_1^k}, z_k) \leq \cdots \leq \kappa(x_{\pi_m^k}, z_k)$ and $\kappa(y_{\sigma_1^k}, z_k) \leq \cdots \leq \kappa(y_{\sigma_n^k}, z_k)$, $\acute{\mathbf{k}}_{X z_k} = [\kappa(x_{\pi_1^k}, z_k), \ldots, \kappa(x_{\pi_m^k}, z_k)]^\top$ and $\acute{\mathbf{k}}_{Y z_k} = [\kappa(y_{\sigma_1^k}, z_k), \ldots, \kappa(y_{\sigma_n^k}, z_k)]^\top$ are the vectors of sorted kernel evaluations, as in the 1D Wasserstein distance. In the case of equal sample sizes $m = n$ and uniform masses $\mu_1 = \cdots = \mu_m = \nu_1 = \cdots = \nu_m = \frac{1}{m}$,

$$MLW_{\kappa,p}^p(\hat{\mu}, \hat{\nu}) = \max_{z \in \mathcal{Z}} W_{\mathfrak{d}_{\phi(z)}^p}(\mu, \nu) = \max_{k \in [l]} \frac{1}{m} \sum_{i=1}^m |\kappa(x_{\pi_i^k}, z_k) - \kappa(y_{\sigma_i^k}, z_k)|^p = \max_{k \in [l]} \frac{1}{m} \|\acute{\mathbf{k}}_{X z_k} - \acute{\mathbf{k}}_{Y z_k}\|_p^p. \tag{7}$$

For general masses or sample sizes, the landmark max-sliced kernel Wasserstein distance can be expressed in terms of the optimal transport plan $\acute{\mathbf{P}}^k$ obtained via the Northwest corner rule for the weights sorted in terms of the $k$th candidate landmark (see Appendix A.2).

For uniform masses $\boldsymbol{\mu} = \frac{1}{m} \mathbf{1}_m$ and $\boldsymbol{\nu} = \frac{1}{n} \mathbf{1}_n$, but possibly non-equal sample sizes $m$ and $n$, the optimal transport solution for all landmarks $\acute{\mathbf{P}}^1 = \cdots = \acute{\mathbf{P}}^{m+n} = \acute{\mathbf{P}}$ is the same and does not depend on the data,

---

[3]This nomenclature contrasts with the concept of a landmark from the Nyström method for kernel matrix approximation, where the landmarks 'guide' the approximation.

which means the Northwest corner rule needs to be computed once. The optimality of a data-independent solution for uniform masses enables efficient computation of the surrogate distribution required for the two-sample permutation test (see Appendix A.3). We note that for $p = 2$ the MLW is lower bounded by the kernel-landmark max-sliced Bures distance (Karahan et al., 2021).

### 3.2 Two-Sample Permutation Test

Given two samples of independent data points $x_1, \ldots, x_m$ and $y_1, \ldots, y_n$ coming from unknown distributions $\mu$ and $\nu$ respectively, we would like to test the null hypothesis $H_0 : \mu = \nu$ compared to the alternative hypothesis $H_1 : \mu \neq \nu$. Given the landmark max-sliced kernel Waserstein distance as a test statistic for comparing two samples, we perform this hypothesis test using a permutation test, creating a surrogate distribution for the test statistic under the null hypothesis of equality of distribution, by permuting data points between the samples (Good, 2013). Calculating the divergence across $B$ random permutations can be performed using a single sort of the columns of the pooled kernel matrix $\mathbf{K}$ to get the sorting permutations $\boldsymbol{\pi}^k$ that sort each column for $k \in [m + n]$ (see Appendix A.3).

Let $D = MLW_{\kappa,p}^p(\hat{\mu}, \hat{\nu})$ denote the original test statistic and $\tilde{D}^b$ denote the test statistic for the $b$th permutation. The p-value is the proportion $\{\tilde{D}^b\}_{b=1}^B$ greater than or equal to $D$, $p_{\text{value}} = \frac{|\{b : \tilde{D}^b \geq D, \quad b \in [B]\}|}{B}$. Given a user-defined significance level $\alpha$, the null hypothesis is rejected if $p_{\text{value}} \leq \alpha$. Across $B$ permutations, the entire procedure requires $\mathcal{O}(n^2 + n^2 \log(n) + Bn^2 + B \log(n)) = \mathcal{O}(n^2 \log(n) + Bn^2)$ operations, which is on the same order as MMD when $\log(n)$ of the same order as $B$, which is the typical case. However, in practice the permutation test can be naively parallelized for both MMD and MLW.

#### 3.2.1 Split Permutation Test for MLW

An alternative approach, which follows prior work on max-sliced kernel Wasserstein distance (Wang et al., 2022), is to split each samples into a training set and a test set. The optimal landmark (and kernel hyper-parameter) is then obtained based on the two training sets, then MLW is calculated on the test sets. In this case, a permutation test involves creating a surrogate distribution under null hypothesis by shuffling the test sets, and does not require identifying a new landmark. Let $z_i^{\text{Tr}}$ and $z_i^{\text{Te}}$ designate the $i$th data points from the pooled training and test samples, respectively, such that $\{z_i^{\text{Tr}}\}_{i=1}^{m_{\text{Tr}}} \cup \{z_i^{\text{Te}}\}_{i=1}^{m_{\text{Te}}} = \{x_i\}_{i=1}^m$ and $\{z_i^{\text{Tr}}\}_{i=1+m_{\text{Tr}}}^{m_{\text{Tr}}+n_{\text{Tr}}} \cup \{z_i^{\text{Te}}\}_{i=1+m_{\text{Te}}}^{m_{\text{Te}}+n_{\text{Te}}} = \{y_i\}_{i=1}^n$. The empirical distributions of the training and test sets are then $\hat{\mu}_{\text{Tr}}$ and $\hat{\nu}_{\text{Tr}}$, and $\hat{\mu}_{\text{Te}}$ and $\hat{\nu}_{\text{Te}}$, respectively. The landmark on the training set and the test statistic are

$$z^* = \arg\max_{z \in \{z_i^{\text{Tr}}\}_{i=1}^{m_{\text{Tr}}}} W_{\mathfrak{d}_{\phi(z)}^p}(\hat{\mu}_{\text{Tr}}, \hat{\nu}_{\text{Tr}}), \quad D_{\text{Te}} = W_{\mathfrak{d}_{\phi(z^*)}^p}(\hat{\mu}_{\text{Te}}, \hat{\nu}_{\text{Te}}). \tag{8}$$

Assuming the split is a fraction of the samples, the computation of the landmark requires $\mathcal{O}(n^2 \log n)$ operations and $\mathcal{O}(n^2)$ memory, but the test statistic requires only $\mathcal{O}(n \log n)$ operations and $\mathcal{O}(n)$ since test points only need to be compared to the landmark. Thus, the cost of the permutation test if $\mathcal{O}(Bn \log n)$, which is less than MMD. However, this split test sacrifices power for this efficiency since a lower sample size is used for the test statistic, the training set only providing the landmark.

## 4 Experimental Results

We detail the results of experiments to compare the ability of MLW ($p = 2$), MMD, and MLB to detect shifts in distribution and interpret these shifts. Quantitative results of the two-sample hypothesis testing are in the form of statistical power tests (the probability of correctly rejecting the null hypothesis). The interpretability of the landmark's witness function is qualitatively examined by showing the instances with the highest witness function evaluations and quantitatively assessed by calculating the proportion of the top instances that are truly associated with known covariate shifts.[4]

---

[4]Note that the class label information is not used by the method; it is only used for evaluation.

### 4.1   MNIST and CIFAR

Following prior work (Rabanser et al., 2019) that used MMD and prior work that used MSKW (Wang et al., 2022) for detecting covariate shift, we create two samples from the images in either MNIST or CIFAR-10 (Krizhevsky, 2009) datasets. The first sample $\hat{\mu}$ is drawn uniformly from the training set, with a uniform distribution across the $C = 10$ classes, while the second sample $\hat{\nu}$, corresponding to the shifted distribution, is drawn by non-uniform sampling across the classes in the test set. Specifically, we consider either downsampling or upsampling one class to create a localized discrepancy. For the downsampling case, let $P_L$ denote the prevalence (expected probability) of the minority class, which is underrepresented in $\hat{\nu}$ compared to the other classes which occur with probability $\frac{1-P_L}{C-1}$. This shift can be introduced by randomly resampling one of the other classes when the $L$th is drawn with probability $p \in [0, 1]$, such that $P_L = \frac{1-p}{C}$. When $p = 0$ $\hat{\nu} = \hat{\mu}$, and the prevalence of all classes is equal at 0.1, but when $p$ is non-zero, $\hat{\nu} \neq \hat{\mu}$. For example when $p$ is 50% (referred to as medium knockout shift (Rabanser et al., 2019), the probability of the underrepresented class is 5%, and the probabilities of the remaining classes are all $0.1056 = 10.56\%$. Thus, the maximal discrepancy in the distributions is localized to the underrepresented digit with a 2-fold decrease, since the prevalence of majority classes has only a 1.056-fold increase.

#### 4.1.1   Data Representations

The kernel-based two-sample hypothesis tests use vectorized image representations, vectors of class probability estimates from a pre-trained classifier as in black box shift detection (BBSD) (Lipton et al., 2018),[5], or embeddings from a pre-trained CLIP vison model `openai/clip-vit-large-patch14-336` (Radford et al., 2021).

The proposed MLW and MMD use a Gaussian kernel $\kappa(x, y) = e^{-\frac{\|x-y\|^2}{2\sigma^2}}$, with the median heuristic, where $\sigma$ is the median of the pairwise distances among all points in the pooled sample. Other kernel-based baseline set $\sigma$ as the median distance between the two samples (Wang et al., 2022)—under the null hypothesis these should be equivalent.

#### 4.1.2   Detecting over-representation in one MNIST Classes (Experiment 1)

We compare MLW's power to the kernel projected Wasserstein distance (KPW) and other baseline methods (Wang et al., 2022), using the published code[6]. MMD-NTK which is a test that uses a neural network trained with the integral probability metric form MMD (Cheng & Xie, 2021). MMD-O, which is just MMD with a log-space optimized Gaussian kernel bandwidth (rather than the median heuristic) (Liu et al., 2020). The interpretable mean-embedding semi-metric that optimizes a kernel-based score function in terms of the locations of multiple landmark points (not constrained to be points) and the Gaussian kernel bandwidth (Jitkrittum et al., 2016). Projected Wasserstein (PW) (Wang et al., 2021) that maximizes the Wasserstein distance in terms of a $d$-dimensional projection ($d = 3$). Training and test splits are 50%-50%. Baseline methods that require hyper-parameters did a further 70% 30% split of the training to optimize any hyper-pameters (Wang et al., 2022). Ten trials of the power test were performed. In each trail, 100 Monte Carlo samples were used to compute power. All baselines used $B = 100$ permutations.

The benchmark is MNIST with upsampling/over-representation of one digit: $\mu$ is uniform across digits and $\nu$ has **increased** prevalence of digit 1, formed by taking a mixture with 85% uniform across classes and 15% for digit 1. The mean and standard-deviation of the statistical power are reported in Table 1. MLW without split has superior performance to all methods. While MMD without split outperforms MLW with split, MLW with split outperforms all previous methods except for KPW at $m = n = 500$. Unlike the other methods, split training and testing is not necessary for MLW and MMD.

Additionally, we report results for the synthetic data that tests the ability of the tests in detecting the performance high-dimensional Gaussians with different covariance in Appendix A.4. We note that the MLW

---

[5]BBSD uses the softmax outputs of a ResNet classifier trained on the entire training set (with validation set used for epoch selection)

[6]`https://github.com/WalterBabyRudin/KPW_Test/tree/main`

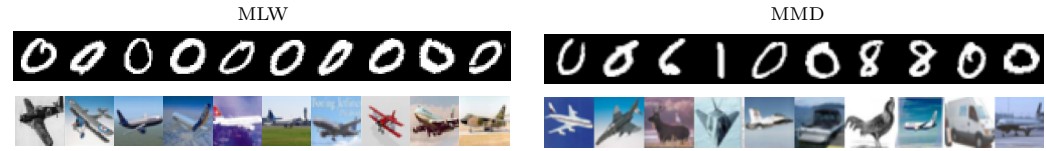

Figure 1: Images from $\hat{\mu}$ with the top-10 largest witness function evaluations using the BBSD learning representation for MLW and MMD. Ideally, the witness function should be largest for instances from the minority class. (Top row) MNIST with minority class '0'. (Bottom row) CIFAR-10 with minority class 'airplane'. The sample sizes are $m, n = 700$; the training set has uniform class distribution, while test set has minority class with prevalence 0.05 (knockout rate of 50%).

without splitting achieves the best performance for some cases of high-dimensional Gaussians with different covariance, the split-version outperforms MMD-based tests MMD-NTK, MMD-O, and ME.

Table 1: Average test power and standard error when upsampling of one class in MNIST. For a sample size, **best is bolded**, *second best*, and third best.

| m=n | MMD-NTK | MMD-O | ME | PW | KPW | MMD | MLW-split | MLW |
|---|---|---|---|---|---|---|---|---|
| 200 | 0.639±0.029 | 0.696±0.006 | 0.298±0.031 | 0.302±0.033 | 0.663±0.015 | *0.944±0.093* | 0.775±0.232 | **0.987±0.033** |
| 250 | 0.763±0.010 | 0.781±0.002 | 0.472±0.017 | 0.369±0.030 | 0.785±0.014 | *0.993±0.013* | 0.911±0.049 | **0.998±0.004** |
| 300 | 0.813±0.016 | 0.869±0.002 | 0.630±0.025 | 0.524±0.023 | 0.928±0.001 | *0.997±0.006* | 0.934±0.033 | **0.999±0.003** |
| 400 | 0.881±0.013 | 0.956±0.003 | 0.779±0.020 | 0.591±0.044 | 0.978±0.000 | **1.000±0.000** | 0.987±0.015 | **1.000±0.000** |
| 500 | 0.950±0.002 | 0.988±0.000 | 0.927±0.006 | 0.782±0.040 | **1.000±0.000** | **1.000±0.000** | 0.996±0.007 | **1.000±0.000** |
| Average | 0.809 | 0.858 | 0.621 | 0.513 | 0.870 | *0.987* | 0.921 | **0.997** |

### 4.1.3 Detecting Imbalanced MNIST and CIFAR Classes (Experiment 2)

Using the framework from prior work Rabanser et al. (2019), we compare MLW and MMD for detecting under-representation of one class on both MNIST and CIFAR-10 dataset across the three learning representations: original vectorized images, which yields meaningful detection performance on MNIST due to the prototypical nature of digits, but much lower statistical power for CIFAR-10; BBSD, which uses the trained classifier representation; and CLIP embeddings, which does not require class knowledge.

First, we show how the witness function, defined by the kernel landmark, is reliably interpretable as an indicator of the specific discrepancies associated with the class imbalance. Namely, we examine if the top-10 instances in the training set with the largest magnitude witness function evaluations are from the test set's minority class. Examples of the instances with the highest witness function evaluations for the MNIST and CIFAR-10 datasets using the BBSD learning representation and MLW and MMD are shown in Fig. 1. Notably, the top-10 witness function evaluations for MLW are all from the minority class, while MMD's witness function evaluations are less precise with 6 out of 10 instances being from the minority class. This is due to the landmark-based witness function being localized by design.

Table 2 summarizes the statistical power results averaged across all classes.

Overall, MLW outperforms MMD on the majority of BBSD cases and CLIP for CIFAR-10.[7] MMD outperforms MLW for MNIST with CLIP embeddings.

On CIFAR-10 across all classes the statistical power for CLIP and BBSD is reported in Table 3, showing that the performance varies by problem even for sample sizes of 2000. Table 4 details the landmark accuracy,

---

[7]Interestingly, in the 'Failing Loudly' framework by Rabanser et al. (2019), MMD on BBSD showed better performance (quantitative results for knock-out on one classs of MNIST and CIFAR-10 each consisted of plots p-values across 5 trials); however, while performing experiments, we noticed that the Gaussian kernel definition is not consistent with the standard definition. Specifically, the implementation uses the kernel $\kappa(x, y) = \exp(-\alpha \|x - y\|^2)$, where $\alpha = \text{median}(\{\|x_i - y_j\|\}_{i=1,j=1}^{m,n})$ is the median distance across pairs between the samples. This mistake means the kernel size $\sigma = \frac{1}{\sqrt{2\alpha}}$ and the resulting kernel is not homogeneous so that the scale of the input matters, which explains its inconsistent performance 'Failing' across learning representations as shown in Table 5 in Appendix A.4.

Table 2: Statistical power for detecting imbalanced distributions on CIFAR-10 using various learning representations (LR) when the test set has a minority class with a prevalence of 5% corresponding to a drop rate of $p = 50\%$. Power is calculated across 100 random draws, hypothesis test use $\alpha = 0.05$ significance level and perform 100 random shuffling to generate the surrogate null distribution. Values are average power across all 10 classes. The best performance per learning representation and sample size is bolded.

| LR | Methods | MNIST | | | CIFAR-10 | | |
|----|---------|-------|---|---|----------|---|---|
| | | Sample size $m = n$ | | | Sample size $m = n$ | | |
| | | 500 | 1000 | 2000 | 500 | 1000 | 2000 |
| Orig | MLW | 0.95 | **1.00** | **1.00** | **0.08** | **0.07** | **0.06** |
| | MMD | **1.00** | **1.00** | **1.00** | 0.02 | 0.03 | **0.06** |
| CLIP | MLW | 0.19 | 0.45 | 0.77 | 0.20 | **0.51** | **0.79** |
| | MMD | **0.70** | **1.00** | **1.00** | 0.17 | 0.30 | 0.77 |
| BBSD | MLW | **0.79** | **0.99** | **1.00** | 0.30 | **0.67** | **0.99** |
| | MMD | 0.10 | 0.12 | 0.11 | **0.35** | 0.16 | 0.48 |

Table 3: Statistical power for detecting imbalanced distributions on CIFAR-10 using various learning representations (LR) when the test set has a minority class with a prevalence of 5% corresponding to a drop rate of $p = 50\%$. Power is calculated across 100 random draws, hypothesis test use $\alpha = 0.05$ significance level and perform 100 random shuffling to generate the surrogate null distribution.

| Sample size | LR | Method | airplane | auto. | bird | cat | deer | dog | frog | horse | ship | truck | average |
|-------------|-----|--------|----------|-------|------|------|------|------|------|-------|------|-------|---------|
| 500 | CLIP | MLW | 0.09 | 0.61 | 0.07 | 0.09 | 0.05 | 0.04 | 0.35 | 0.13 | 0.07 | 0.47 | **0.197** |
| | | MMD | 0.19 | 0.50 | 0.01 | 0.02 | 0.05 | 0.00 | 0.07 | 0.10 | 0.23 | 0.57 | 0.174 |
| | BBSD | MLW | 0.35 | 0.97 | 0.05 | 0.08 | 0.35 | 0.01 | 0.37 | 0.06 | 0.08 | 0.64 | 0.296 |
| | | MMD | 0.46 | 0.86 | 0.09 | 0.16 | 0.42 | 0.08 | 0.47 | 0.15 | 0.25 | 0.6 | **0.354** |
| 1000 | CLIP | MLW | 0.58 | 0.83 | 0.03 | 0.02 | 0.33 | 0.04 | 0.9 | 0.93 | 0.52 | 0.87 | **0.505** |
| | | MMD | 0.39 | 0.56 | 0.12 | 0.00 | 0.18 | 0.01 | 0.27 | 0.24 | 0.52 | 0.75 | 0.304 |
| | BBSD | MLW | 0.74 | 1.00 | 0.49 | 0.38 | 0.75 | 0.05 | 0.89 | 0.64 | 0.88 | 0.92 | **0.674** |
| | | MMD | 0.10 | 0.52 | 0.07 | 0.03 | 0.20 | 0.00 | 0.20 | 0.07 | 0.15 | 0.29 | 0.163 |
| 2000 | CLIP | MLW | 0.97 | 0.77 | 0.44 | 0.49 | 1.00 | 0.26 | 1.00 | 1.00 | 1.00 | 1.00 | **0.793** |
| | | MMD | 0.79 | 0.55 | 0.79 | 0.53 | 0.94 | 0.28 | 0.97 | 0.95 | 0.93 | 0.93 | 0.766 |
| | BBSD | MLW | 1.00 | 0.99 | 1.00 | 0.99 | 1.00 | 0.92 | 1.00 | 1.00 | 1.00 | 1.00 | **0.990** |
| | | MMD | 0.48 | 0.20 | 0.43 | 0.57 | 0.82 | 0.18 | 0.57 | 0.27 | 0.64 | 0.66 | 0.482 |

which is the rate that landmark is in the minority class. Notably, the landmark accuracy is relatively higher for sample sizes of 500 or 1000 compared to statistical power, as shown in Figure 2. For MMD, the precision of the label of the point with highest witness function is around chance (data not shown) across the sample sizes and learning representations.

Table 4: Accuracy of the landmark corresponding to knockout distribution shift on CIFAR-10. After knockout the minority class has a prevalence of 5% compared to the other classes having 10.05%. Accuracy is calculated across 100 random draws.

| Sample size | LR | airplane | auto. | bird | cat | deer | dog | frog | horse | ship | truck | average |
|-------------|-----|----------|-------|------|------|------|------|------|-------|------|-------|---------|
| 500 | | 0.41 | 0.86 | 0.01 | 0.03 | 0.32 | 0.12 | 0.65 | 0.36 | 0.29 | 0.82 | 0.387 |
| 1000 | CLIP | 0.92 | 0.99 | 0.22 | 0.13 | 0.70 | 0.09 | 1.00 | 0.97 | 0.86 | 0.97 | 0.685 |
| 2000 | | 0.99 | 0.99 | 0.87 | 0.81 | 1.00 | 0.59 | 1.00 | 1.00 | 1.00 | 1.00 | 0.925 |
| 500 | | 0.86 | 0.99 | 0.11 | 0.03 | 0.77 | 0.18 | 0.78 | 0.44 | 0.59 | 0.92 | 0.567 |
| 1000 | BBSD | 0.99 | 0.99 | 0.94 | 0.34 | 0.96 | 0.74 | 0.99 | 0.94 | 1.00 | 1.00 | 0.889 |
| 2000 | | 1.00 | 1.00 | 0.99 | 0.35 | 1.00 | 0.99 | 0.99 | 1.00 | 1.00 | 1.00 | 0.932 |

## 5 Discussion

The results indicate that the proposed kernel landmark sliced Wasserstein divergence (MLW) has the ability detect and localize discrepancies between distributions, but performance depends on the representation. As previously noted (Rabanser et al., 2019), the black box shift detector (BBSD) (Lipton et al., 2018) provides the best performance for identifying class imbalance. While we consider fixed kernels on top of

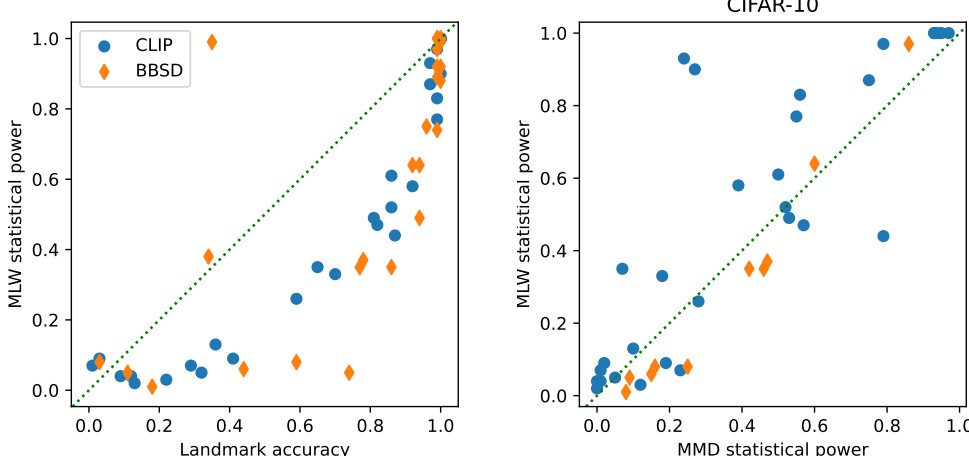

Figure 2: Performance across 10 cases of CIFAR-10 one class knockout (50% drop rate) for CLIP and BBSD learning representations. (Left) The relationship between landmark accuracy and power. (Right) Plots of MLW and MMD statistical power.

fixed representation, following Biggs et al. (2023) a version of MLW that optimizes the kernel function as a combination of base kernel, without the data splitting that lowers performance, has the potential for a more powerful method.

A limitation of MLW is that it selects a single point as a reference point to describe the discrepancy; thus, when multiple discrepancies exists only a single will be identified, leading to an incomplete understanding of distribution shift. Future work should examine algorithms to identify multiple non-redundant landmarks, along the lines of Kim et al. (2016). While the experimental results explored here were limited to cases of synthetically created divergences on standard datasets, future work can explore applications to various domains such as biomedical images.

## 6   Conclusion

The max-sliced kernel landmark Wasserstein distance (MLW) provides a divergence measure that can detect and localize shifts in distributions. The method evaluates the discrepancy between two distributions in terms of the similarity to a landmark point, which for a sample can be computed exactly and efficiently. Through statistical power tests, we show that it is superior to the baseline kernel method maximum mean discrepancy (MMD) in detecting both imbalanced and perturbed distributions on diverse learning representations.

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

# A  Appendix

## A.1  Proofs

*Proof of Theorem 1.* For the subspace $\mathcal{S}$, define the orthogonal projection $\Omega_\mathcal{S} : \mathcal{H} \to \mathcal{S}$. Any $\omega \in \mathcal{F}$ can be written as $\omega_\mathcal{S} + h$, where $h = \omega - \Omega_\mathcal{S}(\omega) \in \mathcal{F}$ and $\langle h, \phi(z_i) \rangle = 0, \quad \forall i \in [m+n]$ via the orthogonal projection. Consequently,

$$W_{\mathfrak{d}_\omega^p}^p(\hat{\mu}, \hat{\nu}) = \inf_{\gamma \in \Gamma(\hat{\mu},\hat{\nu})} \mathbb{E}_{(X,Y)\sim\gamma} \|\omega\|_\mathcal{H} |\omega(X) - \omega(Y)|^p = \inf_{\gamma \in \Gamma(\hat{\mu},\hat{\nu})} \mathbb{E}_{(X,Y)\sim\gamma} \|\omega\|_\mathcal{H} |\omega_\mathcal{S}(X) - \omega_\mathcal{S}(Y)|^p$$
$$\leq \inf_{\gamma \in \Gamma(\hat{\mu},\hat{\nu})} \mathbb{E}_{(X,Y)\sim\gamma} |\omega_\mathcal{S}(X) - \omega_\mathcal{S}(Y)|^p \leq \max_{\omega_\mathcal{S} \in \mathcal{S}} W_{\mathfrak{d}_{\omega_\mathcal{S}}^p}(\hat{\mu}, \hat{\nu}), \tag{9}$$

where the equality follows from $X, Y \in \{z_i\}_{i=1}^{m+n}$ and $\omega(z_i) = \langle \omega_\mathcal{S} + h, \phi(z_i) \rangle = \omega_\mathcal{S}(z_i)$, and the first inequality follows from $\|\omega\| \leq 1$. Taking the supremum over all $\omega \in \mathcal{F}$ yields the equality, $\sup_{\omega \in \mathcal{F}} W_{\mathfrak{d}_\omega^p}^p(\hat{\mu}, \hat{\nu}) = MSKW_{p,\kappa}(\hat{\mu}, \hat{\nu}) = \max_{\omega_\mathcal{S} \in \mathcal{S}} W_{\mathfrak{d}_{\omega_\mathcal{S}}^p}(\hat{\mu}, \hat{\nu})$. $\qquad\square$

*Proof of Theorem 2.* To be a probability distance metric for $\mu, \nu, \xi \in P(\mathcal{X})$ with finite $p$th moments, $MLW_{\kappa,p}$ must satisfy the following (Mueller, 1997)::

- $MLW_{\kappa,p}(\mu, \nu) \geq 0$

- $MLW_{\kappa,p}(\mu, \nu) = MLW_{\kappa,p}(\nu, \mu)$

- $MLW_{\kappa,p}(\mu, \nu) = 0 \iff \mu = \nu$.

- $MLW_{\kappa,p}(\mu, \nu) \leq MLW_{\kappa,p}(\mu, \xi) + MLW_{\kappa,p}(\nu, \xi)$.

Non-negativity and symmetry are obvious from the proprieties of the Wasserstein distance. If $\mu = \nu$, then for any $\omega \in \mathcal{H}$, $\inf_{\gamma \in \Gamma(\mu,\mu)} \mathbb{E}_{(X,Y)\sim\gamma} |\omega(X) - \omega(Y)|^p = 0$.

We assume $\mu \neq \nu$ and proceed to lower bound the distance and show that $MLW_{\kappa,p}(\mu, \nu) > 0$. For any $\omega \in \mathcal{H}$,

$$W_{\mathfrak{d}_\omega^p}^p(\mu, \nu) = \inf_{\gamma \in \Gamma(\mu,\nu)} \mathbb{E}_{(X,Y)\sim\gamma} |\omega(X) - \omega(Y)|^p$$
$$\geq \inf_{\gamma \in \Gamma(\mu,\nu)} |\mathbb{E}_{(X,Y)\sim\gamma}[\omega(X) - \omega(Y)]|^p \tag{10}$$
$$= |\langle \bar{h}(\mu) - \bar{h}(\nu), \omega \rangle|^p, \tag{11}$$

where the inequality follows from Jensen's inequality based on the convexity of $|\cdot|^p$, and the subsequent equality is based on the definition of mean embedding operator $\bar{h}$. Taking the supremum over the set of landmarks yields an expression in terms of the difference of the means in the RKHS

$$\sup_{\omega \in \mathcal{L}} W^p_{\mathfrak{d}^p_\omega}(\mu, \nu) \geq \sup_{\omega \in \mathcal{L}} |\langle \underbrace{\bar{h}(\mu) - \bar{h}(\nu)}_{h_\Delta \in \mathcal{H}}, \omega \rangle|^p \tag{12}$$

$$= \sup_{z \in \mathcal{X}} |\langle h_\Delta, \phi(z) \rangle|^p = \sup_{z \in \mathcal{X}} |h_\Delta(z)|^p. \tag{13}$$

When $\kappa$ is characteristic, the mean embedding operator $\bar{h}(\zeta) : \zeta \mapsto \mathbb{E}_{X \sim \zeta}[\phi(X)]$ is injective for $\zeta \in \mathcal{P}_\mathcal{X}$ (Fukumizu et al., 2008; Gretton et al., 2012), and $\mu \neq \nu \implies \bar{h}(\mu) \neq \bar{h}(\nu) \implies \exists f \in \mathcal{F}, \langle h_\Delta, f \rangle \neq 0$. Equivalently, $\mu \neq \nu \implies \|h_\Delta\|_\infty = \sup_{z \in \mathcal{X}} |h_\Delta(z)| > 0$. Together this yields $MLW_{\kappa,p}(\mu, \nu) \geq \sup_{\omega \in \mathcal{L}} |\langle h_\Delta, \omega \rangle|^p > 0$ for $\mu \neq \nu$.

The triangle inequality follows from the fact that the Wasserstein distance $W_{\mathfrak{d}^p_E}$ is itself is a metric:

$$MLW_{\kappa,p}(\mu, \nu) = \sup_{\omega \in \mathcal{L}} W_{\mathfrak{d}^p_E}(\omega_\sharp \mu, \omega_\sharp \nu)$$

$$\leq \sup_{\omega \in \mathcal{L}} \left( W_{\mathfrak{d}^p_E}(\omega_\sharp \mu, \omega_\sharp \zeta) + W_{\mathfrak{d}^p_E}(\omega_\sharp \nu, \omega_\sharp \zeta) \right) \tag{14}$$

$$\leq \sup_{\omega \in \mathcal{L}} W_{\mathfrak{d}^p_E}(\omega_\sharp \mu, \omega_\sharp \zeta) + \sup_{\omega' \in \mathcal{L}} W_{\mathfrak{d}^p_E}(\omega'_\sharp \nu, \omega'_\sharp \zeta) \tag{15}$$

$$= MLW_{\kappa,p}(\mu, \zeta) + MLW_{\kappa,p}(\nu, \zeta) \tag{16}$$

$$\square$$

## A.2 MLW Computation for General Masses

For general masses or sample sizes, the optimization can be expressed in terms of the non-zero entries of the solution $\acute{\mathbf{P}}^k = \arg\min_{\mathbf{P} \in \mathcal{P}_{\acute{\boldsymbol{\mu}}^k, \acute{\boldsymbol{\nu}}^k}} P_{ij}|i - j|^p$ obtained via the Northwest corner rule for each candidate landmark $k \in [l]$ using the permuted masses $\acute{\boldsymbol{\mu}}^k = [\mu_{\pi^k_i}]_{i=1}^m$ and $\acute{\boldsymbol{\nu}}^k = [\nu_{\sigma^k_i}]_{i=1}^n$,

$$MLW^p_{\kappa,p}(\hat{\mu}, \hat{\nu}) = \max_{k \in [l]} \acute{\mathbf{p}}^\top |(\acute{\mathbf{k}}_{Xz_k})_{\acute{\mathbf{i}}} - (\acute{\mathbf{k}}_{Yz_k})_{\acute{\mathbf{j}}}|^{\circ p}. \tag{17}$$

where $[\acute{p}_t]_{t=1}^s = [\acute{P}_{\acute{i}_t \acute{j}_t}]_{t=1}^s$ are the non-zero entries of $\acute{\mathbf{P}}$ with vectors of row and column indices $\acute{\mathbf{i}}$ and $\acute{\mathbf{j}}$ of length-$s$ ($s$, the cardinality of the support of the transport plan is on the order of $n$).

## A.3 Single-sort Permutation Algorithm

For $b \in [B]$, let $\boldsymbol{\xi}^b \in \mathcal{Q}_{m+n}$ denote the permutation vector that selects $m$ points to form the first surrogate sample $\mathcal{Z}^b_X = \{z_{\xi^b_i}\}_{i=1}^m$ and $n$ points to form the second sample $\mathcal{Z}^b_Y = \{z_{\xi^b_{i+m}}\}_{i=1}^n$. Each column of $\mathbf{K}_{\tilde{X}^b Z} = [\acute{\mathbf{k}}_{\tilde{X}^b z_k}]_{k=1}^{m+n}$ is created as $\acute{\mathbf{k}}^\top_{\tilde{X}^b z_k} = [\kappa(z_{\pi^k_j}, z_k)]_{j \in [m+n]:z_{\pi^k_j} \in \mathcal{Z}^b_X}$ and the remaining entries form $\mathbf{K}_{\tilde{Y}^b} = [\acute{\mathbf{k}}_{\tilde{Y}^b z_k}]_{k=1}^{m+n}$ with entries $\acute{\mathbf{k}}^\top_{\tilde{Y}^b z_k} = [\kappa(z_{\pi^k_j}, z_k)]_{j \in [m+n]:z_{\pi^k_j} \in \mathcal{Z}^b_Y}$. The test statistic for the $b$th permutation is

$$\tilde{D}^b = \max_{k \in [l]} \acute{\mathbf{p}}^\top |\mathbf{R}^X \acute{\mathbf{k}}_{\tilde{X}^b z_k} - \mathbf{R}^Y \acute{\mathbf{k}}_{\tilde{Y}^b z_k}|^{\circ p}, \tag{18}$$

where $\mathbf{R}^X \in \{0, 1\}^{s \times m}$ and $\mathbf{R}^Y \in \{0, 1\}^{s \times n}$ are sparse binary matrices with the $s$ ones at the row-column indices $\{(t, \acute{i}_t)\}_{t=1}^s$ and $\{(t, \acute{j}_t)\}_{t=1}^s$ corresponding to the indexing of rows of the kernel matrices corresponding to $\acute{\mathbf{i}}$ and $\acute{\mathbf{j}}$.

## A.4 Additional Results

Figure 3 illustrates representative computation time for implementations of the samples up to $m = n = 5000$.

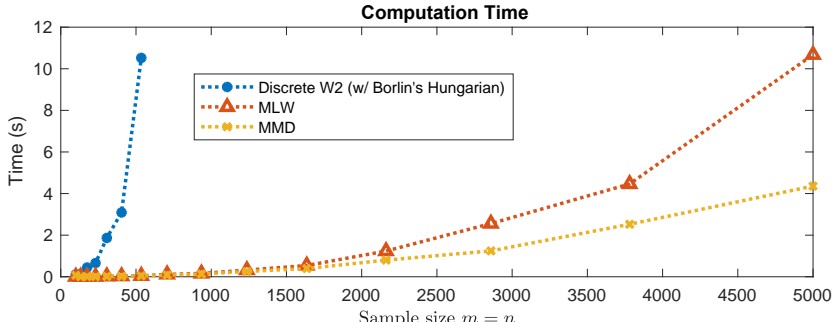

Figure 3: The computation time of kernel Wasserstein-2 distance, MLW, and MMD across different sample sizes $m = n$. The computational complexity of solving the linear program associated with the Wasserstein distance is $\mathcal{O}(n^3)$, MLW is $\mathcal{O}(n^2 \log(n))$, and MMD is $\mathcal{O}(n^2)$. Run time logged in MATLAB 2018b on a MacBook Pro 2.7 GHz Quad-Core Intel Core i7. The linear program is solved as a linear assignment problem using the Hungarian algorithm implemented by Niclas Borlin Carpaneto & Toth (1980).

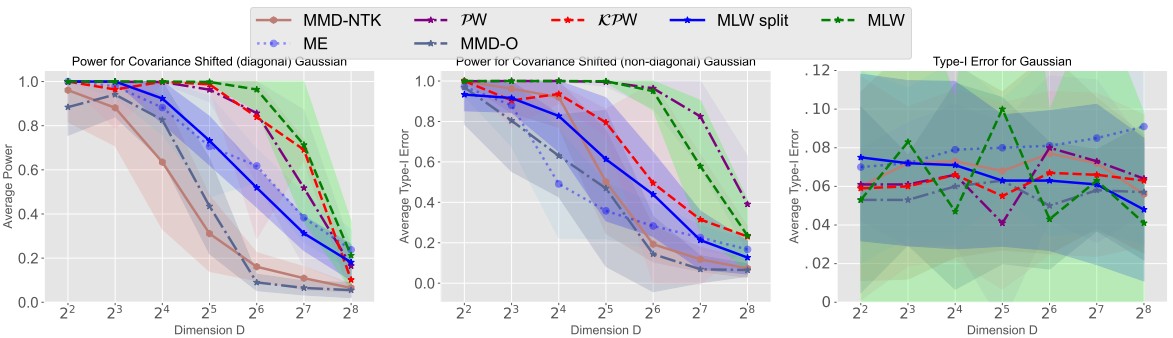

Figure 4: Power curves and Type I error for baselines (Wang et al., 2022) and MLW (split and unsplit) when testing divergence of isotropic Gaussians with different variance in increasing dimension. Tests use $B = 100$ permutations. In each trial, power and error are computed on 100 Monte Carlo generations. Plots are mean across 10 trials with filled area being the 95% confidence interval for Gaussian $\pm 1.95$ standard deviations.

Table 5: Statistical power for detecting imbalanced distributions on CIFAR-10 using various learning representations (LR) when the test set has a minority class with a prevalence of 5% corresponding to a drop rate of $p = 50\%$. Power is calculated across 100 random draws, hypothesis test use $\alpha = 0.05$ significance level and perform 100 random shuffling to generate the surrogate null distribution. Values are average power across all 10 classes. The best performance per learning representation and sample size is bolded. 'Failing' refers to MMD with incorrect kernel size implementation (Rabanser et al., 2019).

| LR | Methods | MNIST Sample size $m = n$ | | | CIFAR-10 Sample size $m = n$ | | |
|----|---------|------|------|------|------|------|------|
| | | 500 | 1000 | 2000 | 500 | 1000 | 2000 |
| Orig | MLW | 0.95 | **1.00** | **1.00** | **0.08** | **0.07** | **0.06** |
| | MMD | **1.00** | **1.00** | **1.00** | 0.02 | 0.03 | **0.06** |
| | Failing | 0.98 | **1.00** | **1.00** | 0.03 | 0.01 | 0.01 |
| CLIP | MLW | 0.19 | 0.45 | 0.77 | 0.20 | **0.51** | **0.79** |
| | MMD | **0.70** | **1.00** | **1.00** | 0.17 | 0.30 | 0.77 |
| | Failing | 0.00 | 0.00 | 0.00 | **1.00** | 0.00 | 0.00 |
| BBSD | MLW | **0.79** | **0.99** | **1.00** | 0.30 | **0.67** | **0.99** |
| | MMD | 0.10 | 0.12 | 0.11 | **0.35** | 0.16 | 0.48 |
| | Failing | 0.18 | 0.43 | 0.92 | 0.29 | 0.298 | 0.918 |

Table 6: Average Type I (false-positive) error with standard deviation for two sample tests using MNIST dataset.

| m=n | MMD-NTK | MMD-O | ME | PW | KPW | MMD | MLW-split | MLW |
|-----|---------|-------|-----|-----|-----|-----|-----------|-----|
| 200 | 0.057±0.010 | 0.056±0.006 | 0.044±0.003 | 0.056±0.004 | 0.061±0.005 | 0.042±0.087 | 0.060±0.017 | 0.030±0.050 |
| 250 | 0.051±0.003 | 0.060±0.001 | 0.065±0.002 | 0.046±0.003 | 0.048±0.002 | 0.041±0.064 | 0.049±0.023 | 0.022±0.019 |
| 300 | 0.068±0.006 | 0.055±0.003 | 0.059±0.007 | 0.056±0.002 | 0.053±0.001 | 0.051±0.044 | 0.063±0.015 | 0.065±0.061 |
| 400 | 0.049±0.007 | 0.058±0.002 | 0.041±0.002 | 0.061±0.006 | 0.056±0.006 | 0.072±0.071 | 0.053±0.031 | 0.055±0.037 |
| 500 | 0.061±0.006 | 0.054±0.004 | 0.060±0.002 | 0.049±0.003 | 0.047±0.004 | 0.035±0.034 | 0.054±0.031 | 0.034±0.030 |
| Average | 0.057 | 0.056 | 0.053 | 0.054 | 0.053 | 0.048 | 0.056 | 0.044 |

