# OpenReview forum: "Finding Landmarks of Covariate Shift with Max-Sliced Kernel Wasserstein Distance"
_TMLR — Rejected by TMLR_

### Review · Reviewer_VAkU · 2025-07-04

**Summary Of Contributions:**

This paper proposes a novel test statistic for detecting covariate shift, called Max-Landmark Wasserstein (MLW). MLW builds on the kernel max-sliced Wasserstein distance, but simplifies the projection step by restricting the witness function to a kernel function centered at a single input point, referred to as the landmark. Instead of optimizing over a broad class of functions in a reproducing kernel Hilbert space (RKHS), MLW selects a single landmark and maps each data point to its kernel similarity with that landmark. This results in one-dimensional representations of the data, over which the Wasserstein distance between the source and target distributions is computed. This formulation yields a test statistic that is both computationally efficient and interpretable, and is particularly effective for identifying localized differences between distributions.

**Audience:**

Yes

**Broader Impact Concerns:**

No such concern.

**Claims And Evidence:**

No

**Requested Changes:**

1. Substantially improve the writing and presentation, addressing the issues mentioned above regarding grammar, undefined terms, inconsistent notation, and reference formatting.
2. Clarify the motivation for MLW more explicitly before presenting the details of it, especially in relation to existing methods like kernel max-sliced Wasserstein distance. What problem does MLW solve that prior methods do not?
3. Elaborate on the optimization procedure used to find the landmark. Specifically, how is this optimization performed, how to ensure that the selected landmark is not a local maximum, what strategies are employed to avoid suboptimal selections (if any), what if the landmark is not (or close) to optimal, does the landmark always exists?
4. Discuss the dependence of MLW on the choice of kernel if possible. How sensitive is the method to kernel hyperparameters? Are certain kernels better suited for specific types of covariate shift?
5. In Experiment 2, the proposed method is only compared to MMD. Why are the baselines from Experiment 1 (e.g., KPW) not included here? A more comprehensive evaluation would strengthen the empirical support for the method.

**Strengths And Weaknesses:**

Strengths:

1. The introduction of a landmark-based slicing mechanism in RKHS for computing the Wasserstein distance is both novel and conceptually appealing. It also enhances interpretability by focusing on local geometric features in the input space.
2. The experiments demonstrate that MLW achieves comparable statistical power to MMD, with the added benefit of interpretability. The results also highlight the choice of data representation may influence the performance.


Weaknesses:

The overall writing of the paper is poor, which significantly impedes readability and comprehension. The manuscript suffers from unclear terminology, grammatical issues, and inconsistent notation and referencing. Several specific issues are outlined below:

1. Undefined or poorly defined terms

- “Slice”: The term is introduced early in the paper (e.g., second paragraph of the Introduction) without a clear explanation. Later, it is referred to as “slice/projection”, which creates ambiguity. Are slice and projection considered interchangeable here?
- “Pushforward measures” and “pushforward distributions”: These important terms are used without being defined. A brief but precise explanation would help readers unfamiliar with optimal transport terminology.
- “Landmark”: The notion of a landmark is central to the method but is not clearly or rigorously defined. The phrase “acting as a slice in RKHS” is especially vague and should be avoided in favor of a more concrete mathematical description.
- “MLB”: This acronym appears without prior definition or explanation.
- Equation (7): It is unclear what the superscript p added on top of MLW denotes. Comparing with Equation (6), it appears to be syntactically redundant or undefined?
- Section 3.2.1: The authors mention “split each sample into a training set and a test set. The optimal landmark is then obtained based on the two training sets…” It is unclear what is meant by “two training sets” in this context.

2. Grammatical and syntactic errors

The paper contains many grammatical errors, including sentence fragments (lacking main verbs) and run-on sentences with multiple main verbs. Examples include:
- Introduction: “Recently, the family of Wasserstein distances, which includes the earth mover’s distances, has seen renewed interest due to its ability to handle provide a meaningful divergence measure…” This sentence contains two main verbs (“handle” and “provide”).
- Section 4.1.2: Four consecutive sentences lack main verbs and are therefore sentence fragments. For example:
“MMD-NTK which is a test that uses a neural network trained with the integral probability metric form MMD… Projected Wasserstein (PW) that maximizes the Wasserstein distance in terms of a d-dimensional projection (d = 3).”
Moreover, the sentence below also contains grammatical issues:
"Unlike the other methods, split training and testing is not necessary for MLW and MMD."

3. Potentially incorrect or inconsistent technical definitions

- In Section 2.3.3, the paper defines the witness function to have norm 1 in RKHS. However, in related work (e.g., Wang et al., 2022, 2024), the RKHS norm constraint is typically ≤1, not exactly equal to 1. This discrepancy should be clarified or corrected.
- The citation to “kernel projected Wasserstein (KPW)” and “kernel max-sliced Wasserstein (KMS)” distances from Wang et al., 2022 and 2024 is inconsistently presented. Perhaps the paper refers to both as “the max-sliced kernel Wasserstein distance (Wang et al., 2022; 2024)”, which does not match the naming in the original works. Additionally, in Section 4.1.2, the baseline is referred to as “kernel projected Wasserstein distance (KPW)” again, which may confuse readers and should be made consistent and faithful to the original terminology.

4. Inconsistent reference formatting

Reference formatting varies significantly, even within citations from the same venue (e.g., NeurIPS). For example:
- “Advances in Neural Information Processing Systems, 29, 2016.”
- “Advances in Neural Information Processing Systems, 36:32539–32573, 2023.”
- “In Advances in Neural Information Processing Systems, volume 20. Curran Associates, Inc., 2008.”

A consistent citation style should be used throughout.

5. Overly long sentences

Several sentences are very long (spanning over 4 lines in some cases), which makes them difficult to parse. It would improve readability to break long sentences into shorter parts if possible.


6. Methodological clarifications needed

The core approach of MLW relies on identifying a single landmark point but the paper lacks an in-depth discussion of:
- The sensitivity of the method to the choice of landmark. How it might affect the robustness of the divergence measure.
- Whether the landmark is guaranteed to exist or be unique.
- The details of the optimization for finding the landmark. How the optimization avoids suboptimal or unstable landmark choices, especially in high-dimensional spaces.

---

### Review · Reviewer_mZ1J · 2025-07-15

**Summary Of Contributions:**

This paper proposes a new kernelized distance measure between two distributions. The main motivation/use is to detect the so-called “covariate shift”, which essentially means the distribution of the test data is different from the training data and one needs to detect this. The methodology of detecting this is to evaluate the distance between the two distributions.

The new measure is called “Landmark Max-Sliced Kernel Wasserstein Distance”. As the name suggests, this is a variant of Wasserstein distance through a kernel. The max-slide method was known before, and the main idea is to choose some “pivot”, which may be a random direction in the feature space, so that the Wasserstein distance is estimated via projecting the dataset to the chosen direction and aggregate. The novelty of the measure in this paper, is a specific way to pick the pivot, which they call “landmark”. Roughly speaking, a landmark is a data point z, and the final Wasserstein distance is evaluated with respect to the kernel-induced distance \sigma(x, y) := |K(z, x) - K(z, y)| where K is the kernel function.

The computational efficiency of this new method is discussed. Experiments on real dataset with artificially planted covariate shift is conducted, and the new method has generally favorable performance than some baselines.

**Audience:**

Yes

**Broader Impact Concerns:**

None.

**Claims And Evidence:**

No

**Requested Changes:**

Apart from the concrete points I mentioned in the “weakness” section, here are further detailed comments.

- The “Theorem 1” looks very confusing — it sounds like your main result, but it does not seem to be.
- In fact, I’m not sure the relation between Theorem 1 and your main result which is the new distance measure. In general, this paper does not present the result/novelty very well.
- Section 2 is very long, and constitutes the most part of your paper. This gives me the impression that your paper is more like a survey than a technical research paper. This may not necessarily be a bad thing, but currently I don’t see how these are useful for presenting your main result, and how your result is related to them.
- Is the landmark idea used anywhere? Do we have any more intuitive motivation of considering this?
- There is only one baseline compared in the experiments. I’m not sure why you only choose this, and how about the many other methods presented in Section 2?

**Strengths And Weaknesses:**

Strength:

- The high-level direction of detecting covariate shift is well motivated
- The new measure is intuitive, and some basic theory around it has been established
- A comprehensive review of relevant definitions is given and I find it useful

Weakness:
- Although many other related distance measures are discussed, it is not immediate how they are related to the new measure. For example, are there any inequalities that you can establish between distance measures? When is it more useful than some other measures (theoretically)? This is also useful for judging the novelty of your method.
- The experiments are not comprehensive enough. One issue is that the datasets are too few/small. Also, the planted shift seems to be somewhat naive, and how does the magnitude of the planted shift affect the performance of your measure is unclear.
- The computational complexity is at least n^2 (I might be wrong), which does not seem to work for big datasets. I would think it’s even worth sacrificing some accuracy for efficiency.

---

### Review · Reviewer_ag9s · 2025-09-13

**Summary Of Contributions:**

The paper proposes a method called using "landmarks" to compute max-sliced kernel Wasserstein distance for two sample test. A witness function is constructed by permuting the samples in kernel space while fixing a single reference sample. The method is useful in computing the max-sliced Wasserstein distance, of which the computation is known to be NP hard. The authors test on an highly unbalanced class examples of MNIST and CIFAR dataset and shows superiority over using the MMD metric.

**Audience:**

Yes

**Claims And Evidence:**

Yes

**Requested Changes:**

1. As mentioned in the weakness, the authors can put more detail in their main analysis part on explaining why the proposed method well-captures the localized changes.

2. Using a table to compare the proposed method with MMD or other methods could be helpful to readers.

3. From Table 2, depending on the representation, MMD seems to show good performance. Can the authors provide more detail on why this is the case?

**Strengths And Weaknesses:**

**Stength**


1. The usage of landmark seems to be an intuitive and simple method to overcome the computational complexity of computing the max-sliced kernel Wasserstein distance. Moreover, the experimental results are positive, which justifies the advantage of the proposed method.

2. The method shows comparable computational complexity to that of MMD.


**Weakness**


1. The paper does not provide a sufficiently clear or detailed explanation of how the proposed method captures localized changes. Although this is presented as a key motivation in the introduction, the concept is only briefly discussed there and is not well-integrated into the main body of the manuscript.

2. The theoretical justification of the usage of this method is not well-established. For example, generalization bounds including Racadmacher complexity result is not provided.

3. The experiments are limited to image-based classification tasks. A simple test on Gausssian model or Higgs UCI dataset could strengthen the paper.

---

### Decision · Action_Editor_EnZS · 2025-10-14

**Recommendation:** Reject

**Additional Comments:**

None

**Audience:**

Yes

**Audience Explanation:**

The problem discussed itself will be interested by TMLR audience.

**Claims And Evidence:**

No

**Claims Explanation:**

As the reviewers pointed out in their review comments in detail, the paper lacks clear justification of the proposed method and many changes were requested. However, these issues were not addressed and thus I cannot recommend its acceptance.